Moonlighting proteins in medically relevant fungi

Arvizu-Rubio Verania J.
García-Carnero Laura C. laura_cgc@hotmail.com
http://orcid.org/0000-0001-6973-0595 Mora-Montes Héctor Manuel hmora@ugto.mx
Department of Biology, Universidad de Guanajuato , Guanajuato, Guanajuato , Mexico
Keller Nancy
Electronic publication date: 2022 Sep 13
Publication date: 2022
Volume: 10
Electronic Location ID: e14001
Received 2022 Jul 11; Accepted 2022 Aug 13
Copyright: © 2022 Arvizu-Rubio et al.
Copyright year: 2022
Copyright holder: Arvizu-Rubio et al.
License: This is an open access article distributed under the terms of the Creative Commons Attribution License, which permits unrestricted use, distribution, reproduction and adaptation in any medium and for any purpose provided that it is properly attributed. For attribution, the original author(s), title, publication source (PeerJ) and either DOI or URL of the article must be cited.
License URL: https://creativecommons.org/licenses/by/4.0/

Keywords: Virulence factor, Cell wall, Alternative function, Host-fungus interaction, Pathogenesis, Adhesin, Chaperone

Funding: Consejo Nacional de Ciencia y Tecnología FC 2015-02-834 Ciencia de Frontera 2019-6380 Red Temática Glicociencia en Salud (CONACYT-México) This work was supported by Consejo Nacional de Ciencia y Tecnología (FC 2015-02-834 and Ciencia de Frontera 2019-6380), and Red Temática Glicociencia en Salud (CONACYT-México). The funders had no role in study design, data collection and analysis, decision to publish, or preparation of the manuscript.

==============================
Moonlighting proteins represent an intriguing area of cell biology, due to their ability to perform two or more unrelated functions in one or many cellular compartments. These proteins have been described in all kingdoms of life and are usually constitutively expressed and conserved proteins with housekeeping functions. Although widely studied in pathogenic bacteria, the information about these proteins in pathogenic fungi is scarce, but there are some reports of their functions in the etiological agents of the main human mycoses, such as Candida spp., Paracoccidioides brasiliensis, Histoplasma capsulatum, Aspergillus fumigatus, Cryptococcus neoformans, and Sporothrix schenckii. In these fungi, most of the described moonlighting proteins are metabolic enzymes, such as enolase and glyceraldehyde-3-phosphate dehydrogenase; chaperones, transcription factors, and redox response proteins, such as peroxiredoxin and catalase, which moonlight at the cell surface and perform virulence-related processes, contributing to immune evasion, adhesions, invasion, and dissemination to host cells and tissues. All moonlighting proteins and their functions described in this review highlight the limited information about this biological aspect in pathogenic fungi, representing this a relevant opportunity area that will contribute to expanding our current knowledge of these organisms’ pathogenesis.

Introduction

Moonlighting proteins are defined as multifunctional proteins with two or more biochemical or biophysical functions, that are not due to gene fusion, splicing, pleiotropic effects, proteolysis, or promiscuous enzymatic activities (Huberts & van der Klei, 2010; Copley, 2012; Amblee & Jeffery, 2015; Jeffery, 2015, 1999). According to the MoonProt database (http://www.moonlightingproteins.org), there are over 500 proteins that have been experimentally identified as moonlighting (Chen et al., 2021), and some of them have been found in pathogenic organisms, playing important roles during the host-pathogen interaction.

Many of the known moonlighting proteins are highly conserved and constitutively expressed housekeeping proteins, found in many organisms and cell types, functioning as chaperones or metabolic enzymes (Gancedo & Flores, 2008; Huberts & van der Klei, 2010; Henderson & Martin, 2011), with alternative functions related to transcription regulation, intron splicing, DNA protection and repair, assembly of organelles, chaperones, adhesion, and cell surface receptors (Gancedo, Flores & Gancedo, 2016). These multitasking proteins may perform their alternative functions at the same time or may change from one to another, representing a switch point among functions that allows the cell to quickly respond to different environmental conditions at a low energetic cost (Jeffery, 2004; Karkowska-Kuleta & Kozik, 2014; Jeffery, 2018). Whether one function or the other is being performed is highly related to (i) the protein cellular localization and cell type, (ii) the available concentrations of the protein partners, (iii) the protein oligomeric state, (iv) post-translational modifications that the protein may undergo, or (v) a combination of these mechanisms (Jeffery, 2018).

A current working model trying to explain moonlighting functions indicates that these are dependent on the ability of the protein to perform them, developed after the canonical function and caused by a small number of mutations with little effect on this first function, which results in improved organism fitness, being therefore enhanced and selected during evolution (Aharoni et al., 2005; Gancedo & Flores, 2008; Huberts & van der Klei, 2010).

Even though many moonlighting proteins and their canonical functions are conserved among organisms, their alternative functions are not, for which it is not possible to predict these second functions using homology, since they do not depend on conserved motifs and domains (Huberts & van der Klei, 2010; Copley, 2012). However, it has been observed that many of these proteins, with cytoplasmic and extracellular functions, share certain structural and biophysical characteristics in their primary amino acid sequences and tertiary structures, such as the lack of signal peptide and anchors or motifs that attach them to the cell surface, high aliphatic indexes, high molecular weights, acidic isoelectric points, and negative hydropathic scores, features usually observed in cytosolic proteins (Amblee & Jeffery, 2015; Jeffery, 2018). An increasing number of moonlighting proteins have been identified as intrinsically disordered proteins or with intrinsically disordered domains or regions (IDDs) often related to the interaction of the protein with one or many partners, due to disorder-to-order transitions and alternative conformations that provide them with structural malleability (Jeffery, 2015; Liu & Jeffery, 2020; Tompa, Szász & Buday, 2005). Therefore, moonlighting functions can derive from modifications of the protein sequence, (i) which can be large conformational changes or transitions between IDDs and multiple distinct folded structures, so that different protein structure conformations can perform different functions, (ii) or can be subtle changes in small portions of amino acids in the protein sequence needed for the second function without altering the protein structure or affecting the region responsible for the canonical function (Copley, 2012; Jeffery, 2018).

Targeting of moonlighting proteins to the cell surface is still poorly understood, since they lack conventional secretion signals, such as the signal peptide, but it has been well proven that the finding of these proteins at the surface is due to secretion and not to cell leakage or adsorption from surrounding dead cells (Jeffery, 2018). Therefore, secretion systems in charge of targeting cytosolic moonlighting proteins to the surface are capable of distinguishing those that should stay inside the cell to perform their canonical functions from proteins that should be extracellularly located (Jeffery, 2018; Copley, 2012). Some non-conventional secretory pathways have been suggested for the transport of these and other proteins that lack signal peptides, including (i) passive protein transfer through the plasma membrane, (ii) membrane flipping with the transfer of attached cytoplasmic proteins to the surface, (iii) translocation via soluble or membrane-bound transporters (Karkowska-Kuleta & Kozik, 2014), (iv) secretory vesicles or endosomal sub-compartments engaged in secretion (Karkowska-Kuleta & Kozik, 2014), (v) exosome-like vesicles (Rodrigues & Djordjevic, 2012), (vi) secretory lysosomes (Nickel & Rabouille, 2009), (vii) and post-translational translocation into the endoplasmic reticulum to enter the secretory pathway (Hernández-Chávez et al., 2014; Schmoll et al., 2016).

Over the last few years, moonlighting proteins have been shown to play important roles not only in the maintenance of basic cellular functions but also during the host-pathogen interaction, representing an important and wide set of virulence factors that enhance the pathogen’s ability to cause infection.

Moonlighting proteins as fungal virulence factors

Although the role of moonlighting proteins during the pathogen-host interaction is widely reported for bacteria (Henderson & Martin, 2011), this is a phenomenon less explored in fungi, probably due to the difficulty to study protein function in eukaryotic pathogens. However, some moonlighting proteins participating during fungal infection have been suggested and confirmed in pathogenic fungi, such as Candida spp., Paracoccidioides brasiliensis, Histoplasma capsulatum, Aspergillus fumigatus, Cryptococcus neoformans, and Sporothrix schenckii.

The fungal cell wall, and the capsule in capsulated fungal species, is the first structure that encounters the host cells and molecules during infection, and participates in the pathogen adherence to the host tissues and interaction with the host immune effectors. This structure is composed of polysaccharides, pigments, and proteins, which work as virulence determinants or factors during infection (Díaz-Jiménez et al., 2012; Mora-Montes et al., 2009; Teixeira et al., 2014; Tóth et al., 2019; Gómez-Gaviria & Mora-Montes, 2020; Navarro-Arias et al., 2019). Some of the cell wall proteins have been reported to be conserved intracellular proteins moonlighting at the cell surface, with housekeeping canonical functions related to the following (Satala et al., 2020a): (a) Metabolic pathways, including (i) glycolysis and gluconeogenesis: fructose-bisphosphate aldolase (Fba), phosphoglycerate mutase (Gpm), glyceraldehyde-3-phosphate dehydrogenase (GAPDH or Tdh3), phosphoglycerate kinase (Pgk), glucose-6-phosphate isomerase 1 (Gpi1), triosephosphate isomerase (Tpi), glycerol-3-phosphate dehydrogenase 2 (Gpd2), enolase (Eno), trehalose-6-phosphate phosphatase (Tsl), and fructose-1,6-bisphosphatase (Fbp); (ii) fermentation: alcohol dehydrogenase (Adh); (iii) pentose phosphate pathway: 6-phosphogluconate dehydrogenase (Gnd), transketolase, and transaldolase; and (iv) Krebs and glyoxylate cycles: malate synthase.

(b) Protein synthesis, that include (i) ribosomal proteins, (ii) elongation factors: elongation factor 2 (Eft2), transcription elongation factor (Tef1); and (iii) chaperones: Ssa1 protein, Ssa2 protein, and heat shock proteins 60 (Hsp60) and 70 (Hsp70).

(c) Redox homeostasis: peroxiredoxin (Tsa), and peroxisomal catalase (Cta).

Most of the moonlighting proteins reported in pathogenic fungi working as virulence factors have been found on the cell surface, associated with the cell wall, or secreted. Thus, discrimination between proteins that have been indeed transported to the cell surface unconventionally and those that come from cell lysis or experimental artifacts needs to be done, to confirm their moonlighting status. Several methods have been used for this, which include protein immunolocalization at the cell wall in intact cells by flow cytometry (Long et al., 2003; Stie, Bruni & Fox, 2009; Luo et al., 2013; Dasari et al., 2019; Gozalbo et al., 1998; Gil-Navarro et al., 1997), and electron and fluorescence microscopy (Gozalbo et al., 1998; Long et al., 2003; Barbosa et al., 2006; Nogueira et al., 2010; Silveira et al., 2013; da Silva Jde et al., 2013; Crowe et al., 2003; Angiolella et al., 1996); surface protein biotinylation and precipitation (López-Ribot et al., 1996; Long et al., 2003; Lopez et al., 2014; Karkowska-Kuleta et al., 2011; Kozik et al., 2015; Karkowska-Kuleta et al., 2016); isolation of the cell wall or cell wall components generating intact protoplasts (Gil-Navarro et al., 1997; Pitarch et al., 2002; Li et al., 2003; Castillo et al., 2008; Longo et al., 2014; García-Carnero et al., 2021); and isolation of extracellular vesicles (Vallejo et al., 2012). As mentioned, the panorama of moonlighting proteins in fungi is limited when compared to bacteria, and a significant amount of information we currently have is on non-pathogenic fungal species, such as Saccharomyces cerevisiae (Gancedo, Flores & Gancedo, 2016; Flores & Gancedo, 2011; Gancedo & Flores, 2008; Rodríguez-Saavedra et al., 2021). The information about moonlighting proteins in fungal pathogens of medical relevance is currently limited and searches in bibliography repositories indicate that the newest review manuscript on this subject was published in 2014 (Karkowska-Kuleta & Kozik, 2014), with a broad vision of eukaryotic pathogens, and when considering particular pathogenic species, the most recent review report is from 2020, gathering information about moonlighting proteins in the C. albicans cell wall (Satala et al., 2020a). Thus, a focused bibliographic search for moonlighting proteins in medically relevant fungal species has not been reported previously, and this review manuscript offers a thorough bibliographic revision on this subject (Table 1). Because of the reason above mentioned, this information will be of interest to the audience specialized in fungal biology, medical mycology, and medical microbiology. In a broader scope, it will be also of interest to the audience working in the cell biology and biochemistry fields, where moonlighting proteins are relevant too.

Table 1 Moonlighting proteins in medically relevant fungal species.

Protein	Canonical function	Moonlighting function	References	
Candida spp.	
GAPDH	Glycolysis and gluconeogenesis	Binding to fibronectin, laminin, and plasminogen	Gozalbo et al. (1998), Crowe et al. (2003)	
Eno1	Glycolysis and gluconeogenesis	Binding to plasmin, plasminogen, fibronectin, vitronectin, laminin, and kininogen
Binding to medical devices	Jong et al. (2003), Kozik et al. (2015)	
Gpm1	Glycolysis and gluconeogenesis	Binding to plasminogen, vitronectin, FH, FHL-1, and kininogen	Poltermann et al. (2007), Lopez et al. (2014)	
Gpd2	Glycolysis and gluconeogenesis	Binding to plasminogen, FH, and FHL-1	Luo et al. (2013)	
Adh1	Fermentation	Binding to plasminogen	Crowe et al. (2003)	
Cta1	Redox homeostasis	Binding to plasminogen	Crowe et al. (2003)	
Tef1	Elongation factor	Binding to plasminogen	Crowe et al. (2003), Poltermann et al. (2007)	
Fba1 and Pgk1	Glycolysis and gluconeogenesis	Binding to plasminogen
Binding to medical devices	Crowe et al. (2003); Poltermann et al. (2007)	
Tsa1	Redox homeostasis	Binding to plasminogen, and kininogen	Crowe et al. (2003)	
Eft2	Elongation factor	Binding to kininogen	Seweryn et al. (2015), Kozik et al. (2015)	
Tpi1	Glycolysis and gluconeogenesis	Bind to kininogen, vitronectin, fibronectin, collagen, laminin, and elastin	Seweryn et al. (2015), Satala et al. (2021)	
Gpi1	Glycolysis and gluconeogenesis	Binding to kininogen, fibronectin, vitronectin, and laminin	Seweryn et al. (2015), Kozik et al. (2015)	
Gnd1	Pentose phosphate pathway	Binding to kininogen, fibronectin, vitronectin, and laminin	Seweryn et al. (2015), Kozik et al. (2015)	
Fbp	Glycolysis and gluconeogenesis	Bind to fibronectin, vitronectin, and laminin	Kozik et al. (2015)	
Malate synthase	Krebs and glyoxylate cycles	Bind to fibronectin, vitronectin, and laminin	Kozik et al. (2015)	
Transketolase and transaldolase	Pentose phosphate pathway	Bind to fibronectin, vitronectin, and laminin	Kozik et al. (2015)	
Ssa1	Chaperone	Binding to endothelial N-cadherin, epithelial E-cadherin, and histatine 5	Sun et al. (2010), Li et al. (2003)	
Ssa2	Chaperone	Binding to idem	Sun et al. (2008), Li et al. (2003)	
Als3	Adhesin	Ferritin receptor	Almeida et al. (2008)	
Paracoccidioides brasiliensis	
Eno	Glycolysis and gluconeogenesis	Binding to laminin, fibronectin, plasminogen, and collagen type I and IV	Donofrio et al., 2009, Nogueira et al. (2010)	
Tpi	Glycolysis and gluconeogenesis	Binding to laminin	Pereira et al. (2007)	
Malate synthase	Krebs and glyoxylate cycles	Binding to fibronectin, and collagen type I and IV
Interaction with Eno and Tpi	da Silva Neto et al. (2009), de Oliveira et al. (2013)	
GADPH	Glycolysis and gluconeogenesis	Binding to laminin, fibronectin, and collagen type I	Barbosa et al. (2006)	
Fba	Glycolysis and gluconeogenesis	Binding to plasminogen
Interaction with macrophages	Chaves et al. (2015)	
14-3-3 protein	Regulation of many vital processes	Binding to laminin, and fibronectin	Andreotti et al. (2005), Marcos et al. (2016)	
Histoplasma capsulatum	
HIS-62	Chaperone	Binding to macrophage CR3 (CD11/CD18)	Long et al. (2003)	
Aspergillus fumigatus	
Eno	Glycolysis and gluconeogenesis	Binding to plasminogen, FH, FHL-1, and C4BP
Allergen	Dasari et al. (2019)	
Tsl	Trehalose biosynthesis	Chitin synthase regulation	Thammahong et al. (2017)	
Cryptococcus neoformans	
Hsp70	Chaperone	Macrophage and monocyte interaction and activation through the CD14 receptor
Plasminogen binding	Asea et al. (2000), Silveira et al. (2013)	
Pgk, Fba, and pyruvate kinase	Glycolysis and gluconeogenesis	Binding to plasminogen	Stie, Bruni & Fox (2009)	
Hsp60	Chaperone	Binding to plasminogen	Stie, Bruni & Fox (2009)	
Transaldolase	Pentose phosphate pathway	Binding to plasminogen	Stie, Bruni & Fox (2009)	
ATP-synthase alpha and beta subunits	Energy transduction	Binding to plasminogen	Stie, Bruni & Fox (2009)	
Response to stress-related protein	Stress signaling pathways	Binding to plasminogen	Stie, Bruni & Fox (2009)	
Glutamate dehydrogenase	Glutamate metabolism	Binding to plasminogen	Stie, Bruni & Fox (2009)	
Sporothrix schenckii	
Hsp60	Chaperone	Binding to laminin, elastin, fibrinogen, and fibronectin	García-Carnero et al. (2021)	
Note:

GAPDH, glyceraldehyde-3-phosphate dehydrogenase; Eno, enolase; Gpm, phosphoglycerate mutase; Gpd2, glycerol-3-phosphate dehydrogenase 2; Adh, alcohol dehydrogenase; Cta, peroxisomal catalase; Tef1, transcription elongation factor; Fba, fructose-bisphosphate aldolase; Pgk, phosphoglycerate kinase; Tsa, peroxiredoxin; Eft2, elongation factor 2; Tpi, triosephosphate isomerase; Gpi1, glucose-6-phosphate isomerase 1; Gnd, 6-phosphogluconate dehydrogenase; Fbp, fructose-1,6-bisphosphatase; Ssa1 and Ssa2: Hsp70, Als3: cell wall agglutinin-like sequence protein 3, HIS-60: Hsp60, Tsl, trehalose-6-phosphate phosphatase.

Survey methodology

The PubMed (https://pubmed.ncbi.nlm.nih.gov/) and Google Scholar (https://scholar.google.com/) repositories were used to search for the following terms together with relevant Boolean Operators and MeSH terms identified for individual databases: moonlighting protein, moonlighting protein and fungi, moonlighting protein and fung*, fungal cell wall and moonlighting protein, cell wall and moonlighting, Candida and moonlighting, Paraccoccidioides and moonlighting, Histoplasma and moonlighting, Aspergillus and moonlighting, Cryptococcus and moonlighting, Sporothrix and moonlighting, yeast and moonlighting protein, hypha and moonlighting protein, filament cell and moonlighting protein, atypical cell wall protein, multifunctional protein and fung*, multitasking protein and fung*, and noncanonical cell wall protein.

Candida spp

Candida species are opportunistic fungal pathogens that most frequently establish the infective process in immunocompromised individuals, causing superficial and systemic infections, and representing an important health threat worldwide (Spampinato & Leonardi, 2013; Martínez-Duncker, Díaz-Jímenez & Mora-Montes, 2014). The main reported species causing infection is Candida albicans, but recently, the incidence of other Candida species has increased, and these include Candida glabrata, Candida tropicalis, Candida parapsilosis, Candida krusei, Candida guilliermondii, Candida auris, and Candida lusitaniae (Gómez-Gaviria & Mora-Montes, 2020; Tóth et al., 2019; Turner & Butler, 2014; Pappas et al., 2010). For the establishment of candidiasis, both pathogen and host-related factors are required and important. Many Candida virulence factors and determinants have been already described, including the expression of adhesins and secreted hydrolytic enzymes, dimorphism, and biofilm formation (Ciurea et al., 2020). The Candida cell wall contains canonical wall proteins found in other yeast-like organisms, but proteomic analyses have revealed the presence of cytosolic proteins in the cell wall (Pitarch et al., 2002; Castillo et al., 2008; Hernáez et al., 2010; Gil-Bona et al., 2018), which can be classified according to their canonical functions into (i) enzymes involved in metabolic pathways, (ii) factors associated with protein synthesis, (iii) chaperones, (iv) and redox homeostasis enzymes (Satala et al., 2020a). Many of these proteins are still considered putative moonlighting proteins, but for some of them, the alternative function has been experimentally confirmed (Table 1 and Fig. 1).

Figure 1 Schematic representation of fungal moonlight proteins localization and function.

Fungal moonlight proteins related to virulence are mostly adhesins with their moonlight function in the cell surface and their canonical function in the cytosol. An asterisk (*) indicates that Als3 and Tsl are moonlight proteins that perform both functions, canonical and moonlight, in the same cellular compartment.

In C. albicans, GAPDH protein found at the cell surface, especially on blastoconidia, participates in the adhesion to fibronectin, laminin, and plasminogen (Gozalbo et al., 1998; Crowe et al., 2003), contributing to fungal adhesion, dissemination, and damage to host tissue. The cell wall-associated GAPDH protein represents about 20–35% of the total amount of this protein in the cell and has been reported to be an important immunogenic antigen that elicits a humoral immune response (Gil-Navarro et al., 1997; Seidler, 2013). However, this protein does not seem to be a suitable therapeutic target against candidiasis, since active immunization with the recombinant protein, opsonization of yeasts with anti-GAPDH antibodies, and passive transfer of polyclonal anti-GAPDH antibodies failed to generate a protective immune response in the host (Gil et al., 2006).

Eno1 has been characterized as a cell surface plasmin(ogen)-binding protein that enhances fibrinolysis and the fungal ability to cross the blood-brain barrier in vitro, participating in tissue invasion and dissemination (Jong et al., 2003). Deletion of its gene impairs C. albicans growth on glucose-containing media, alters hyphal formation, increases drug susceptibility, and eliminates the fungal ability to kill the host (Ko et al., 2013). This protein binds to the cell wall glucans, being tightly attached to the wall’s inner layer (Angiolella et al., 1996). Although Eno1 is an immunodominant antigen that stimulates cellular and humoral responses during infection (Sundstrom, Jensen & Balish, 1994), its recombinant version is only capable of providing modest protection against candidiasis (Montagnoli et al., 2004). Intracellularly, this protein has been reported to present two enzymatic activities given by two different catalytic sites, as enolase, which is essential for the glycolytic and gluconeogenesis pathways (Satala et al., 2020b), and as transglutaminase, participating in fungal growth and morphogenesis, cell division and osmotic regulation (Reyna-Beltrán et al., 2018). Blocking of Eno1 transglutaminase activity, alone or in combination with fluconazole, provided an antifungal activity against C. albicans in vitro and in vivo, proposing this protein as a putative therapeutic target (Li et al., 2019). In addition, human Eno has also been found anchored on the cell surface of human cells, where it binds to plasminogen and represents a local fibrinolysis regulator (Miles et al., 1991). This moonlight function seems to be conserved among organisms, and is has been proven, at least in humans and some pathogenic organisms, to be due to the presence of lysine residues on the C-terminal end of the protein, which appear to function as plasminogen and extracellular matrix (ECM) components binding sites (Miles et al., 1991; Bergmann et al., 2003; Derbise et al., 2004; Vanegas et al., 2007; Rahi et al., 2018; Satala et al., 2020a).

The C. albicans alcohol dehydrogenase 1 (Adh1) catalyzes the conversion of acetaldehyde to ethanol and is also involved in the NADH-dependent methylglyoxal dehydrogenase activity that generates pyruvate (Kwak, Ku & Kang, 2014). The adh1Δ null mutants showed an increment in the methylglyoxal levels, growth defects, and virulence attenuation (Kwak, Ku & Kang, 2014).

Gpm1 protein is found on the C. albicans yeast and hypha cell surface, where it binds to plasminogen, vitronectin, and the host complement regulators Factor H (FH) and Factor H-like binding protein (FHL-1), facilitating attachment to endothelial cells and keratinocytes, degradation of the host ECM, and promoting immune evasion (Poltermann et al., 2007; Lopez et al., 2014). GMP1 gene deletion decreased fungal binding to endothelial cells, confirming the importance of the encoded protein for C. albicans virulence (Lopez et al., 2014). Also, C. albicans secreted Gpd2 protein constitutes a plasminogen-, FH-, and FHL-1-binding protein (Luo et al., 2013).

Eno1 protein, GAPDH protein, and Gpm1 protein, along with the moonlighting proteins Adh1, Tsa1, and Cta1 represent the major plasminogen-binding proteins, contributing to 85% of the plasminogen binding in C. albicans cell wall protein extracts (Crowe et al., 2003). Tef1 protein, Pgk1 protein, and Fba1 protein also participate in plasminogen binding, though to a lesser extent (Crowe et al., 2003; Poltermann et al., 2007).

C. albicans Eno1 protein, Gmp1 protein, Gpi1 protein, Eft2 protein, Tpi1 protein, Tsa1 protein, and Gnd1 protein are atypical cell wall proteins and have all been identified as kininogen-binding proteins (Karkowska-Kuleta et al., 2011; Seweryn et al., 2015). Kininogen is an important protein in a proteolytic cascade of human plasma that participates in the host pro-inflammatory, antimicrobial, and anti-adhesive responses (Lalmanach et al., 2010). C. albicans and C. glabrata Tpi1 protein has also been proven to bind to vitronectin, fibronectin, collagen, laminin, and elastin, suggesting its contribution to fungal adhesion (Satala et al., 2021).

C. parapsilosis and C. tropicalis pseudohyphae also have cell surface moonlighting adhesins that include malate synthase, Gpi1 protein, Gnd1 protein, Eno1 protein, Fbp protein, transketolase, transaldolase, and Eft2 protein, which bind to fibronectin, vitronectin, and laminin (Kozik et al., 2015). In addition, C. tropicalis Eno1 protein and Gpm1 protein, and C. parapsilosis Gnd1 protein also bind to high-molecular-mass kininogen (Karkowska-Kuleta et al., 2016; Karkowska-Kuleta et al., 2017).

These moonlighting proteins not only mediate the binding of Candida spp. cells to host components but also to medical devices, which is an important trait during biofilm formation. Eno1 was found to participate in the adhesion of C. albicans, C. glabrata, C. krusei, and C. parapsilosis to plastic surfaces; Fba1 protein favors the adhesion of C. albicans, C. glabrata, and C. krusei; and Pgk1 protein is involved in the adhesion of C. parapsilosis to different components of medical devices (Núñez-Beltrán, López-Romero & Cuéllar-Cruz, 2017).

Ssa1, a member of the Hsp70 family, has been localized throughout the C. albicans cell wall (López-Ribot et al., 1996), where it binds to endothelial cells N-cadherin and epithelial cells E-cadherin, thus being essential for host cell endocytosis and invasion (Sun et al., 2010). SSA1 gene deletion decreased C. albicans adhesion, damage, and invasion of the epithelium and endothelium, resulting in lower virulence (Sun et al., 2010), confirming the important role of this protein during infection. However, Ssa1 and Ssa2 proteins, another Hsp70, are reported to also have a detrimental effect on the fungus, due to their ability to bind to histatine 5, a salivary histidine-rich antifungal peptide, that causes cell death when adsorbed to the cell surface (Sun et al., 2008; Li et al., 2003, 2006). Ssa1 and Ssa2 proteins are considered the main immunogenic antigens in the C. albicans cell wall and are capable of eliciting cell-mediated and humoral immune responses (Martínez et al., 1998; López-Ribot et al., 2004). However, vaccination with the recombinant proteins failed to protect against candidiasis, but instead enhanced the infection (Bromuro et al., 1998).

Als3 protein, a member of the C. albicans cell wall agglutinin-like sequence (Als) family (Hoyer et al., 1998), is an important multifunctional protein in the hyphae and pseudohyphae surface (Hoyer et al., 1998; Argimón et al., 2007). Als3 has a canonical function of adhesin and invasin that binds to gelatin, fibronectin, fibrinogen, type IV collagen, laminin, salivary pellicle, and E- and N-cadherin, thus mediating adherence and internalization to endothelial and epithelial cells (Sheppard et al., 2004; Phan et al., 2007; Nobbs, Vickerman & Jenkinson, 2010), and also plays an important role in biofilm formation (Nobile et al., 2006; Zhao et al., 2006). However, Als3 protein has an additional function as a ferritin receptor in C. albicans hyphae, where it binds to purified ferritin and ferritin contained within epithelial cells, helping the fungus to obtain iron from the host (Almeida et al., 2008). ALS3 gene deletion decreased fungal adherence to endothelial and oral epithelial cells (Zhao et al., 2004), decreased endocytosis by oral epithelial cells and vascular endothelial cells in vitro, altered biofilm structures (Nobile et al., 2006; Zhao et al., 2006), and diminished iron acquisition in vitro (Almeida et al., 2008), demonstrating its multiple functions and their importance during infection. Als3 protein is also an immunodominant antigen, and vaccination with its recombinant version protected against vaginal and disseminated candidiasis (Spellberg et al., 2006).

Paracoccidioides brasiliensis

The thermodimorphic fungi P. brasiliensis and Paracoccidioides lutzii are the etiological agents of paracoccidioidomycosis, an important human systemic disease with multiple clinical forms, endemic to Latin America (San-Blas, Niño-Vega & Iturriaga, 2002; Mendes et al., 2017). The interaction of these fungi with their host and their successful colonization depends on many regulatory mechanisms and virulence factors that allow them to cause infection, which include morphogenesis, changes in cell wall polysaccharide composition, adhesion, adaptation to environmental stresses, and production of extracellular vesicles (de Oliveira et al., 2015; Camacho & Niño-Vega, 2017).

Proteomic analyses of P. brasiliensis yeasts and mycelium cell walls, and vesicle and vesicle-free extracellular content have revealed the presence of many cytoplasmic proteins, mostly related to carbohydrate and protein metabolism, stress response, oxidation/reduction, translation, nucleic acid binding, and cellular architecture (Table 1) (Vallejo et al., 2012; Longo et al., 2014). Also, proteomic analysis of the Paracoccidioides secretome identified 15 plasminogen binding proteins, many of which are cytoplasmic proteins (Chaves et al., 2015). Out of these proteins, some have already been proven to be moonlighting proteins that participate as adhesins during infection (de Oliveira et al., 2015; Marcos et al., 2014). Eno protein, Fba protein, GADPH protein, Tpi protein, and malate synthase have been found in the Paracoccidioides cell wall and extracellular vesicles, mediating adhesion to the host tissues (Vallejo et al., 2012). Eno protein was found to be highly expressed when P. brasiliensis grows in the presence of blood and has the ability to bind to laminin, fibronectin, plasminogen, and type I and IV collagens (Donofrio et al., 2009; Nogueira et al., 2010; Marcos et al., 2012), representing 80% of fungal adhesion to epithelial cells (Donofrio et al., 2009), which demonstrates its essential role in fungal attachment, internalization, and invasion to the host tissues. Tpi protein is a yeast cytosolic and cell wall antigen that reacts with sera of patients with paracoccidioidomycosis and that participates in fungal binding to epithelial cells, with affinity to laminin (da Fonseca et al., 2001; Pereira et al., 2007). Malate synthase is also a P. brasiliensis cell wall and secreted protein that functions as an adhesin that binds to fibronectin and type I and IV collagens (da Silva Neto et al., 2009). In addition, this protein interacts with other cell wall adhesins, such as Eno protein and Tpi protein, enhancing the fungal adhesion (de Oliveira et al., 2013). GAPDH protein was found in P. brasiliensis extracellular vesicles and cell wall, with increased expression in the yeast morphology and during the mycelium-yeast transition (Barbosa et al., 2006, 2004; Longo et al., 2014). On the cell surface, this protein binds to laminin, fibronectin, and type-I collagen, being involved in the fungal adhesion to pneumocytes during the early stages of infection (Barbosa et al., 2006). Fab protein binds to plasminogen and participates in the fungal interaction with macrophages, increasing fibrin degradation and internalization by phagocytes (Chaves et al., 2015).

A cytosolic 30 kDa glycoprotein, found highly expressed in the P. brasiliensis cell wall during infection and in high virulent strains, was identified as a member of the 14-3-3 protein family (da Silva Jde et al., 2013; Andreotti et al., 2005) of conserved regulatory eukaryotic proteins involved in transcription, signal transduction, protein localization and degradation, cell cycle, apoptosis, and many other processes (Sluchanko & Gusev, 2017). The cell surface 14-3-3 protein is an important Paracoccidioides adhesin that binds to laminin and fibronectin (Andreotti et al., 2005; Marcos et al., 2016), and the silencing of its gene affected the yeast morphology, impaired morphological switching, and decreased laminin binding, resulting in a significant virulence reduction (Marcos et al., 2016), underlining its importance to the P. brasiliensis virulence.

Histoplasma capsulatum

H. capsulatum is an environmental thermodimorphic fungus with two pathogenic varieties, H. capsulatum var capsulatum, and H. capsulatum var duboissi, that cause human histoplasmosis (Antinori, 2014), a worldwide distributed pulmonary and systemic disease with a mortality rate up to 7% (Kauffman, 2007; Armstrong et al., 2018). Disease development and severity depend on the host immune status, virulence of the fungal strain, and the fungal load inoculated (Knox & Hage, 2010). Only a handful of H. capsulatum virulence factors have been identified, and these include the presence of α-1,3-glucan at the cell surface, thermotolerance, melanin, and surface ligands for immune receptors (Mihu & Nosanchuk, 2012). Thus far, only one of these virulence factors has been proven to be a moonlighting protein (Table 1).

At the first stage of histoplasmosis infection, H. capsulatum gets recognized and internalized by alveolar macrophages through the integrin receptor CR3 (CD11/CD18). Once inside, this fungus impairs macrophage activation and phagolysosomal fusion, and replicates, killing the alveolar macrophages and invading neighboring and recruited macrophages, thus disseminating to other organs from the lungs (Bullock & Wright, 1987; Newman et al., 1990). This defective macrophage response against H. capsulatum depends, in part, on the receptor that the fungus uses to enter phagocytes, and Hsp60, also known as HIS-62, has been reported to be the major H. capsulatum surface protein that interacts with CD11/CD18 (Long et al., 2003). This intracellular chaperone has been found moonlighting on the outer surface of the H. capsulatum cell wall, and although HIS-62 protein lacks the classical secretion signal, it shows a unique leader sequence upstream of the amino terminus of the mature protein, with high homology to leader sequences of other pathogenic fungi (Long et al., 2003), which might be related to its secretion. Also, the cell wall-associated Hsp60 was found to be N-linked glycosylated, while the intracellular protein is not, suggesting that the surface HIS-62 protein enters the secretory pathway (Long et al., 2003).

Hsp60 has been reported as an immunogenic antigen during histoplasmosis, and antibodies against this protein provided a protective immune response in the host, by reducing the fungal burden and increasing phagolysosomal fusion in vitro (Guimarães et al., 2009). Also, vaccination with the native and recombinant HIS-62 protected against pulmonary and systemic histoplasmosis, which suggests that this protein is an important target for the cellular and humoral immune response (Gomez, Gomez & Deepe, 1991; Gomez, Allendoerfer & Deepe, 1995).

Aspergillus fumigatus

Aspergillus species are environmental molds found worldwide (Hope, Walsh & Denning, 2005). A. fumigatus is the most common species associated with aspergillosis syndromes (Park & Mehrad, 2009), which range from allergic noninvasive clinical forms in immunocompetent patients to systemic disease under immunosuppression conditions, with mortality rates up to 100% if the diagnosis is missed or delayed (Brown et al., 2012; Latgé & Chamilos, 2019; Barnes & Marr, 2006). The Aspergillus spp. pathogenicity is dependent on the host immune response and pulmonary microbiome, and on the presence of fungal virulence factors that might be in part enhanced by some host effectors (Raksha, Singh & Urhekar, 2017; Kolwijck & van de Veerdonk, 2014). Different A. fumigatus virulence factors have been identified, which include adhesins, pigments, hydrolytic enzymes, catalases and superoxide dismutases, mycotoxins and non-protein metabolites, thermotolerance, allergens, and zinc acquisition (Raksha, Singh & Urhekar, 2017; Szalewski et al., 2018).

One of the main A. fumigatus adhesin and allergen is the moonlighting protein Eno (Table 1), expressed at the swollen conidia and hypha cell surface, where it binds to plasminogen and represents a ligand for FH, FHL-1, and C4BP (C4 binding protein) (Dasari et al., 2019). The affinity of this protein to the plasma complement regulators contributes to the fungal immune evasion, since its binding to FH and FHL-1, and C4PB inactivates C3b and C4b, respectively, while binding to plasminogen degrades C3 and C3b. Thus, Eno protein contributes to cell attachment and invasion, tissue damage, and immune evasion (Dasari et al., 2019). Eno protein is also an important Aspergillus allergen that causes a hypersensitive immune reaction in the host and is highly recognized by IgE (Lai et al., 2002; Gautam et al., 2007; Shah & Panjabi, 2014).

The A. fumigatus Tsl protein was identified as an important enzyme for trehalose biosynthesis and cell wall structure. This protein directly interacts with chitin synthase and regulates its activity and localization, as a moonlight function, affecting the cell wall integrity and structure (Thammahong et al., 2017). Deletion of its encoding gene reduced trehalose content in conidia and mycelium but also altered the cell wall integrity, by increasing chitin, β-glucan, and galactosaminogalactan content and exposure (Thammahong et al., 2017). Also, these mutants trigger a much stronger pro-inflammatory response in the host, probably due to the exposure of those cell wall components at the cell surface, but without affecting the fungal virulence (Thammahong et al., 2017).

Other pathogenic fungi

The encapsulated yeast C. neoformans is an opportunistic fungus that can cause cryptococcal meningoencephalitis in immunocompromised patients (Brown et al., 2012). Macrophages play an important role in fungal killing, but also in the replication and dissemination of C. neoformans (García-Rodas & Zaragoza, 2012), for which fungal surface components that interact and attach to phagocytes are essential for C. neoformans pathogenesis. Hsp70 has been detected on the C. neoformans cell surface where it is involved in macrophage and monocyte interaction and activation, through the CD14 receptor (Table 1) (Asea et al., 2000; Silveira et al., 2013). C. neoformans Hsp70 inhibits macrophage activation, thus helping the fungus to evade killing by the phagolysosome oxidative response, and also promotes attachment to epithelial cells through a still unidentified receptor that might be shared with glucuronoxylomannan, the main capsular antigen (Silveira et al., 2013). In addition, many cytosolic proteins with plasminogen affinity have been identified on the C. neoformans cell surface, including Pgk protein, Fba protein, Hsp60, Hsp70, transaldolase, pyruvate kinase, ATP-synthase alpha and beta subunits, the response to stress-related protein, and glutamate dehydrogenase (Stie, Bruni & Fox, 2009).

The thermodimorphic fungal species of the Sporothrix pathogenic clade are the causative agents of sporotrichosis, a human and animal subcutaneous mycosis that can disseminate and cause systemic infections (Lopes-Bezerra et al., 2018; Lopez-Romero et al., 2011; Mora-Montes et al., 2015; Nava-Pérez et al., 2022). Sporothrix schenckii and Sporothrix brasiliensis are the most commonly isolated species and also the most studied, but only a few of their virulence factors and determinants have been identified, which include adhesins, peptidorhamnomannan, morphological switching, and melanin (Tamez-Castrellon et al., 2020; García-Carnero & Martínez-Álvarez, 2022). Some intracellular proteins, such as Hsp60, Hsp70, Tef1 protein, mitochondrial peroxiredoxin, lipase 1, Eno1 protein, and pyruvate kinase, have been found in cell wall preparations from S. schenckii cells growing in the presence of different stressors (Ruiz-Baca et al., 2019), however, because the preparations were performed using whole-cell homogenates, it remains to be confirmed whether these are indeed proteins found on the S. schenckii surface with moonlighting functions. Proteomic analysis of the S. schenckii yeast-like cells’ peptidorhamnomannan revealed that it is composed of 325 possible proteins, many of which are intracellular proteins with housekeeping functions and without conventional signal sequences, that may be moonlighting proteins related to virulence on the cell surface (García-Carnero et al., 2021). One of the main peptidorhamnomannan proteins is Hsp60, which was proven to function as a cell surface adhesin with binding affinity to laminin, elastin, fibrinogen, and fibronectin, being essential for fungal virulence (Table 1) (García-Carnero et al., 2021). Also, this protein appears to be highly immunogenic, and immunization with its recombinant version or treatment with anti-Hsp60 antibodies can protect laboratory animals against a lethal infection with S. schenckii (García-Carnero et al., 2021).

The host perspective

It is widely known that the adherence of a pathogen to its host represents an essential step to invade, colonize, evade the immune response, and cause infection, as already mentioned. Most of the moonlighting proteins that participate in fungal virulence processes functions as adhesins that bind to important components of the host ECM and of homeostatic and immune cascades (Fig. 2).

Figure 2 Schematic representation of the host ligands that bind to the different fungal moonlighting proteins.

The host ligands are located in many cell types and compartments, and many of the pathogen moonlighting proteins are capable of binding to more than one ligand, facilitating the attachment and dissemination of the fungus to different tissues. All the proteins are present on the cell surface of the pathogenic fungus, which is arbitrarily represented by yeast cells. GAPDH, glyceraldehyde-3-phosphate dehydrogenase; Eno, enolase; Gpm, phosphoglycerate mutase; Gpd2, glycerol-3-phosphate dehydrogenase 2; Adh, alcohol dehydrogenase; Cta, peroxisomal catalase; Tef1, transcription elongation factor; Fba, fructose-bisphosphate aldolase; Pgk, phosphoglycerate kinase; Tsa, peroxiredoxin; Eft2, elongation factor 2; Tpi, triosephosphate isomerase; Gpi1, glucose-6-phosphate isomerase 1; Gnd, 6-phosphogluconate dehydrogenase; Fbp, fructose-1, 6-bisphosphatase; Ssa1 and Ssa2: Hsp70, Als3: cell wall agglutinin-like sequence protein 3, HIS-60: Hsp60.

The ECM is a dynamic network of macromolecules that provides structural support for cells and tissues, composed mainly of proteoglycans and glycosaminoglycans, elastin and elastic fibers, collagens, laminins, fibronectin, thrombospondin, and vitronectin (Karamanos et al., 2021). Through signal transduction, the ECM regulates many cellular functions, that include growth, proliferation, cell migration, and differentiation, playing an essential role in cell homeostasis (Karamanos et al., 2021). Therefore, disruption of any of the ECM components alters tissue function and regeneration, thus facilitating infection by pathogenic fungi (Tomlin & Piccinini, 2018). Disruption of the ECM during infection can be caused directly by the degradation of its components after attachment through the fungus proteases, elastases, or collagenases; or indirectly through the host catabolic machinery that gets altered upon infection or by the activity of activated macrophages that destroy the ECM (Tomlin & Piccinini, 2018). In addition, fungal binding to plasminogen causes its cleavage to plasmin, an important protease of the fibrinolytic system that degrades fibrin (Castellino & Ploplis, 2005), which participates in blood coagulation, cell migration, and tissue repair; activates collagenases that degrade collagens and activates complement mediators; and degrades several ECM components, such as fibronectin, laminin, and thrombospondin, thus degrading the host cell barriers and ECM (Singh et al., 2012). Finally, the ECM also participates in the host immune response, by providing important signaling to the immune cells for their proliferation, differentiation, activation, and migration, which gets altered with the ECM disruption (Tomlin & Piccinini, 2018).

High molecular weight kininogen is a member of the human contact system, that upon activation cleaves kininogen and releases antimicrobial peptides and bradykinin, involved in the regulation of inflammatory processes, vascular permeability, and blood pressure (Ponczek, 2021). The binding of pathogenic fungi to kininogens through different ligands triggers inflammatory responses through bradykinin, resulting in vascular leakage, vasodilatation, and increased blood flow, enabling the inflow of plasma nutrients, and enhancing pathogen dissemination and invasion (Karkowska-Kuleta & Kozik, 2014; Oehmcke-Hecht & Köhler, 2018).

The complement system is an important effector of the innate immune response, that participates in the elimination of pathogenic organisms through proinflammatory responses, which include attraction and activation of phagocytes, and enhanced opsonization (Friese et al., 1999). The plasma proteins FH and FHL-1 are regulatory elements that control the alternative complement activation at the level of C3b, an active complement-derived product, and the binding of pathogenic fungi to these regulators mediates immune evasion and disruption of the complement activation (Behnsen et al., 2008), thus promoting invasion and dissemination.

Conclusions

Although long known, moonlighting proteins have received little attention when participating in virulence processes of pathogenic fungi. To date, many moonlighting proteins have been suggested to function as fungal virulence factors, but only a few have been experimentally confirmed, with very limited information about their alternative functions, the mechanisms they use to perform them, and their cellular localization and trafficking. The information that we know about these moonlighting functions during infection highlights the importance these proteins have in fungal pathogenesis, by enabling fungal attachment, invasion, and dissemination in the host, along with helping with the fungal immune evasion. Therefore, they are key factors for understanding fundamental and clinically relevant biological processes in these organisms. Even though their study should be a priority, it has been neglected. C. albicans is perhaps the most studied species in medical mycology, and therefore, most of the available information about moonlighting proteins has been generated on this organism. However, as we know, fungal infections are not reduced to only candidemia/candidiasis, other fungal pathogens are relevant in this field. As reported in this review article, the information about moonlighting proteins in other fungal species is limited when compared with that generated in Candida species, stressing that more efforts are required to understand the contribution of these proteins to fungal pathogenesis, in a broader sense.

These proteins represent an example of genetic economy and directed evolution, being a low-cost adaptation mechanism for the cell when facing stressful conditions, such as those generated in the host during infection, and at the same time, contributing to the fungal virulence. As reviewed in this text, a great percentage of the reported fungal moonlighting proteins are conserved metabolic enzymes and chaperones, with alternative functions related to adhesion, tissue damage, and immunoevasion. The moonlighting function may likely have an impact on other processes during the interaction with the host, such as cell wall remodeling, posttranscriptional regulatory events, apoptosis, and cancer development, as reported for bacterial moonlighting proteins (Podobnik et al., 2009; Banerjee et al., 2007; Lin et al., 2010; Basak et al., 2005). Thus, this represents an area of opportunity and a challenge in the years to come.

Additional Information and Declarations

Competing Interests

Author Contributions

Data Availability

Héctor M. Mora-Montes is an Academic Editor for PeerJ.

Verania J. Arvizu-Rubio performed the experiments, analyzed the data, prepared figures and/or tables, authored or reviewed drafts of the article, and approved the final draft.

Laura C. García-Carnero conceived and designed the experiments, performed the experiments, analyzed the data, prepared figures and/or tables, authored or reviewed drafts of the article, and approved the final draft.

Héctor Manuel Mora-Montes conceived and designed the experiments, analyzed the data, authored or reviewed drafts of the article, and approved the final draft.

The following information was supplied regarding data availability:

This is a review article.

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
