# Peer review of "Moonlighting proteins in medically relevant fungi"

_PeerJ, doi:10.7717/peerj.14001_

## Round 0.1 · original submission · Minor Revisions

We have had 3 experts review your submission and all of them have good suggestions which I would ask you to respond and pay attention too. There will be quite a bit of editing but still, just editing.

Reviewer 1 ·

Basic reporting

The review is of broad and cross-disciplinary interest and within the scope of the journal. It covers a subset of moonlighting proteins that has not been so thoroughly reviewed recently so it helps bring together examples that some of the readers might not have previously known about. The introduction helps explain the importance of the topic of moonlighting proteins and gives an overview of some recent discoveries and developments in studies of microorganisms. The last few sentences of the introduction could be rephrased to make this clearer.
I have many suggestions for improving the flow of the manuscript and word choices. The main concern is a large number of run-on sentences. The authors should cut many or the sentences into two (usually where a comma is located). Use of gene names and protein names should be more consistent, and if the gene name is used, the protein name and/or catalytic function should be indicated. I plan to attach a PDF with notes. Addition of a column containing one or two references for each protein in the Table 1 would be helpful. One or more figures would be a nice addition.

Experimental design

The Survey Methodology is consistent with a comprehensive, unbiased coverage of the subject and sources are adequately cited for the most part. One additional reference should be the one where moonlighting proteins was first defined (Jeffery CJ. Moonlighting proteins. Trends Biochem Sci. 1999 Jan;24(1):8-11. doi: 10.1016/s0968-0004(98)01335-8. PMID: 10087914.). The review is organized logically into paragraphs and subsections.

Validity of the findings

The conclusions bring the topic back to the main idea and identify some gaps in our knowledge and general future directions.

Annotated reviews are not available for download in order to protect the identity of reviewers who chose to remain anonymous.

Reviewer 2 ·

Basic reporting

In the submitted manuscript Authors took up an interesting issue concerning moonlighting proteins in selected fungal species. As more and more information is available on this group of proteins and it is quite up-to-date topic, the summary is a valuable contribution to the current knowledge about fungal pathogenesis. This work is nice and written properly. However I have some comments on the manuscript. One question is what is actually new in this review to the previous ones relating to moonlighting proteins in fungi? Species other than Candida were reported in the review several years ago, but the work on Candida species is fairly recent. Moreover, some parts of this manuscript quite closely resemble those previous ones, especially in the introductory part. Please highlight the novelty and justification of this review and make sure that all work on moonlighting proteins since previous reviews is covered.

Experimental design

The name “moonlighting protein” is not always assigned to every protein of this type, some of them are referred in the literature to as atypical cell wall proteins, multifunctional, multitasking proteins etc. hence it would be necessary to broaden the search, even for specific protein names.

Validity of the findings

The main aim of the review is poorly defined, whether it is just to list examples of moonlighting proteins in various species - then Table 1 largely coincides with the text and is a repetition of information from the second part of the manuscript. One would expect a broader approach to the problem. Authors often mention the immunogenicity of individual proteins and their potential to be used in the prevention of fungal infections - I propose to include a separate subsection on this topic including different proteins, with in-depth discussion. There is no emphasis on the general importance of the phenomenon of protein moonlighting in relation to fungal pathogenesis. Specific host proteins that are ligands for fungal proteins are mentioned repeatedly i.e. plasminogen, kininogen, ECM proteins. Authors could extend the work to a more detailed discussion of the problem from the host perspective, how moonlighting proteins of different species modulate fungal interactions with individual host systems and are involved in the development of inflammation, fibrinolysis, etc. It would be valuable to try to see from the perspective of all the species of fungi discussed, as some proteins are shared between them, is there any pattern in these interactions? Perhaps an additional schematic drawing of the host's site would be helpful.
Referring to the line 119-123, does the way the protein got to the surface determine whether or not the protein is moonlighting? Rather its different function and a different location, please clarify this issue.
The fragment about the Als3 protein is confusing, it is a protein important in pathogenesis, located on the surface, having many functions, but not considered a moonlighting protein, because its additional functions are not significantly different from the primary one, they complement each other in the same location. I advise to remove this fragment as it does not correspond to the subject of the review.
Moreover, is there a certainty that some moonlighting proteins do not share structural host protein binding motifs? There are some studies pointing to specific fragments in enolases that play a role in plasminogen binding. Authors should compare examples from different species and discuss the issue.

Additional comments

In a Table 1, it is additionally necessary to include appropriate references next to the examples.

Reviewer 3 ·

Basic reporting

Yes, the review is of broad and interdisciplinary interest, and it is within the scope of the journal.
Yes, the field has been reviewed recently, and the information generated will be helpful for those interested in opportunistic fungal pathogens of humans.
The Introduction does adequately introduce the topic and makes it clear who is the audience and the objectives of this review.

Experimental design

Yes, the Methodology used is consistent with comprehensive coverage of the fungal Moonlighting Proteins (MPs). Sources are appropriately cited. However, some references are suggested to discuss the atypical enzymatic activity of some Moonlighting proteins like Eno1.
The review is organized in a logical and organized manner. The authors only present a Table, suggestions were made to improve Table 1. In addition, the incorporation of a Figure that integrates all the functions and characteristics of the Moonlighting proteins of the different fungal species is suggested.

Validity of the findings

Yes, there is a well-developed and well-supported argument that meets the objectives set out in the Introduction.
However, in the comments made to this review, it is suggested that the Eno1 enzyme of C. albicans be discussed in a particular way since, in addition to presenting a location in the wall as a Moonlighting protein, it presents an atypical enzymatic activity, due to a second catalytic domain. Therefore, it will be necessary to discuss it to coincide with the conclusions. Future directions stemming from this review are very clear, MPs are vital factors in understanding the fundamental and clinically relevant biological processes in fungal pathogens.

Additional comments

Comments Moonlight Proteins (MP) PeerJ
1. Line 47
What low cost are you referring to? Energetic? To be more punctual.

2. Line 52,
In what type of nucleotide sequences are these mutations found? On which motifs, domains, or the active site? What do they affect: binding to the substrate or the cofactor? Or do they affect the protein's secondary, tertiary, or quaternary structure? To the secretion pathway? This paragraph is very general; could the authors please mention some examples.

3. Line 98
No more fungal species?

4. Almost all of the examples of C. albicans MPs in Table 1 are proteins that bind to other proteins or to host components.
In point 10, it is suggested to mention proteins with atypical enzymatic activity, such as Eno 1.
Other non-canonical activities, for example, protection against stress, molecular mimicry, etc., can be mentioned in Table 1.

5. Line 242
What does the following paragraph refer to?

“Despite all the examples already described, not all multipurpose proteins involved in virulence are proteins with a canonical housekeeping function.
C. albicans”

6. Line 296
Discuss a little more about the functions of the 14-3-3 family of proteins

7. Lines 393-399
Could a comparison be made between the proteins found in the cell wall preparations of S. schenckii by Ruiz-Baca in 2019 and García-Carnero in 2021 and mention them in Table 1 as putative multi-use proteins?

8. From what can be seen in Table 1, an excellent compilation of Moonlight proteins was made in different species of pathogenic fungi. Nevertheless,
to make the information in Table 1 more understandable, the abbreviations and acronyms of the enzyme names should be repeated in a legend at the end of the table. For example glyceraldehyde-3-phosphate dehydrogenase (GAPDH) and ensure that when it is first mentioned in the main text, all acronyms are defined.
In addition, it is suggested to include an additional column in Table 1, with the corresponding References for each MPs of each species of fungus.
9. It is suggested to include a Figure that integrates all the information generated in this review. An informative, integrating and updated figure, involving all the fungal MPs of the different species mentioned in this review. It could be a Figure equivalent to Fig. 2 of the review by Satala et al., 2020, but more complete and updated.

10. Overall, this review is very informative and will undoubtedly guide those studying fungal MPs and their role. However, the most frequently mentioned aspect is the moonlighting function of these proteins due to their different subcellular locations in this review. In most studies, it has been reported that the moonlighting functions of intracellular proteins of C. albicans in the cell surface are not enzymatic. However, the multifunctionality of proteins can also include multiple catalytic activities, which can lead to another essential aspect worth discussing: atypical enzyme activity.
In this sense, there are some very well-documented examples in C. albicans:

a) Eno1 of C. albicans

After studying the inhibition of enzyme activities with anti-CaEno1 antibodies and through bioinformatic studies, the authors suggested that the catalytic sites of enolase and transglutaminase are located in different protein domains.

Reference:
Reyna-Beltran, E., Iranzo, M., Calderon-Gonzalez, K. G., Mondragon-Flores, R., Labra-Barrios, M. L., Mormeneo, S., et al. (2018). The Candida albicans ENO1 gene encodes a transglutaminase involved in growth, cell division, morphogenesis, and osmotic protection. J. Biol. Chemistry 293, 4304–4323. doi: 10.1074/jbc.M117.810440.

b) On the other hand, it has been shown that the inhibition of transglutaminase activity in C. albicans in synergy with fluconazole affects the dimorphism and growth of C. albicans, proposing Eno1 as a therapeutic target for the design of antifungals.

Reference:
Li L, Zhang T, Xu J, Wu J, Wang Y, Qiu X, Zhang Y, Hou W, Yan L, An M, Jiang Y. The Synergism of the Small Molecule ENOblock and Fluconazole Against Fluconazole-Resistant Candida albicans. Front Microbiol. September 6, 2019; 10:2071. doi: 10.3389/fmicb.2019.02071. PMID: 31555252; PMID: PMC6742966.

c) Adh1 is another example in C. albicans of proteins with dual enzymatic activity.
Adh1 which catalyzes the conversion of acetaldehyde to ethanol and is also involved in the NADH-dependent catalysis of the oxidation and reduction of methylglyoxal (MG).

Reference:
Kwak, MK; Ku, M.; Kang, SO NAD+-linked alcohol dehydrogenase 1 regulates methylglyoxal concentration in Candida albicans. FEBS Lett. 2014, 588, 1144–1153, doi:10.1016/j.febslet.2014.02.042.

Annotated reviews are not available for download in order to protect the identity of reviewers who chose to remain anonymous.

---

## Round 0.2 · accepted · Accept

Your reply has adequately addressed the concerns of the reviewers.